# Changes Connected to Early Chronic Pancreatitis and Early Pancreatic Cancer in Endoscopic Ultrasonography (EUS): Clinical Implications

**DOI:** 10.3390/cancers17111891

**Published:** 2025-06-05

**Authors:** Natalia Pawelec, Łukasz Durko, Ewa Małecka-Wojciesko

**Affiliations:** Department of Digestive Tract Diseases, Medical University of Lodz, 90-153 Lodz, Poland; natalia.pawelec@stud.umed.lodz.pl (N.P.); lukasz.durko@umed.lodz.pl (Ł.D.)

**Keywords:** ECP, early PDAC, EUS

## Abstract

Detecting chronic pancreatitis and pancreatic cancer at an early stage is crucial in diagnosing these conditions. This may contribute to a better prognosis and earlier implementation of effective therapy. Among the diagnostic methods, EUS shows the best results in diagnosing these changes.

## 1. Introduction

Chronic pancreatitis and pancreatic cancer are associated with dangerous complications and poor prognosis. EUS, CT scans, magnetic resonance imaging, and US are used for pancreatic imaging. The pancreas is difficult to visualize due to its small size and retroperitoneal localization. Chronic pancreatitis and pancreatic cancer are associated with severe outcomes, so early detection and treatment could be beneficial. However, early changes typical of pancreatic disease are very subtle and not always visible with imaging techniques. In addition, PDAC laboratory biomarkers, like CA 19-9, are not sensitive and specific enough, and they can be elevated in other types of tumors and benign pathologies [1,2,3].

## 2. Chronic Pancreatitis

Chronic pancreatitis (CP) is defined as an irreversible, progressive inflammatory disease [4]. Pancreatitis accounts for more than 200,000 hospitalizations in the United States each year [5]. In Europe, it is estimated that CP affects 5–10 people per 100,000 inhabitants [6]. It is characterized by morphologic changes that are associated with pain, and eventually, impairment of both exocrine and endocrine functions [7]. CP is connected to a reduction in average life expectancy (by 10 years) and an increased incidence of PDAC. The risk of pancreatic cancer in patients with CP is up to 14.7–25 times higher than in the general population [8,9]. Inhibition of pancreatic inflammation progression may theoretically protect against advanced disease as well as the development of pancreatic cancer [10].

Diagnosis of late CP is easy, even with clinical history and basic imaging techniques. On the other hand, the diagnosis of early chronic pancreatitis is challenging due to the lack of sensitive and specific methods [11]. The current diagnosis is based on clinical examination and imaging techniques. Pancreatic enzymes may be mildly elevated or normal [12]. A fecal elastase-1 test level of <200 μg/g indicates exocrine insufficiency with mediocre efficacy. Nevertheless, according to some authors, pancreatic exocrine insufficiency may belong to early CP (ECP) symptoms [13,14]. Unfortunately, this test has very low sensitivity in mild CP and about 75% in moderate and severe disease [8]. In addition, this test has a high false-positive rate.

CT is a first-choice modality, but it can mostly detect large duct disease (pancreatic ductal dilatation of 7 mm or more) [12].

EUS and US-guided pancreatic biopsy specimens for CP diagnosis are not always relevant, because inflammatory changes are not uniformly dispersed within the gland. In addition, this is an invasive method and may even be complicated by acute pancreatitis [12].

## 3. Imaging Modalities for Chronic Pancreatitis

The majority of CP and PDAC diagnoses are based on imaging methods, such as transabdominal US, CT, MRI, ERCP, and EUS [1].

### 3.1. Transabdominal Ultrasonography (US)

The early stage of CP is characterized by minimal morphological changes, which are not visible on the US. This technique has limitations, such as dependence on operator experience and blurred pancreatic structure, particularly in obese patients. Even after adding contrast, such as SonoVue (sulfur hexafluoride), ECP may appear similar to a healthy pancreas [6]. Furthermore, bowel gas, obesity, and individual variations make it difficult to detect early changes [1]. On the other hand, the US is useful in imaging and monitoring CP complications, such as pseudocysts and pseudoaneurysms [6].

### 3.2. Computed Tomography (CT)

CT is widely accessible and shows a detailed view of pancreatic and adjacent organ morphology [15]. In addition, CT can rule out other intra-abdominal diseases, upper gastrointestinal cancers, and pathologies that may cause similar symptoms [1,15,16]. It can also identify complications of acute and chronic pancreatitis, such as pseudocysts, biliary or duodenal obstruction, venous thrombosis, and pseudoaneurysms [15].

Ductal changes, such as dilation, strictures, contour irregularity, and parenchymal or intraductal calcification, represent the characteristic image of CP on CT. If at least two of the following features are visible—main pancreatic duct (MPD) within 2–4 mm; mild organ enlargement; irregular main pancreatic duct with ≥3 pathological side branches; pseudocysts ≤ 10 mm; and heterogeneous parenchyma—they are proposed by experts as signs of early chronic pancreatitis (ECP) [17].

There are limited studies assessing the usefulness of CT in the detection of ECP. Minimal changes typical of ECP, which are associated with the pancreatic duct, are difficult to detect on CT [15]. The sensitivity and specificity for CP detection on CT were 74–90% and 85–91%, respectively [18,19]. Figure 1 presents imaging of ECP in CT.

### 3.3. Magnetic Resonance Imaging (MRI)

MRI and magnetic resonance cholangiopancreatography (MRCP) are recommended in patients with no specific changes detected on CT but with a strong suspicion of chronic pancreatitis (CP) [1]. MRCP allows for accurate visualization of the biliary tract and pancreatic ducts. It provides imaging of static and slowly moving fluids [16]. Although the technique is precise, it does not always detect subtle changes in the main pancreatic duct (MPD) and its side branches [15]. For example, branch ducts narrower than 1 mm are not visible with this technique. On the other hand, dilatation, strictures, and irregularities of the MPD can be demonstrated by MRCP [20].

In early chronic pancreatitis (ECP), MRI may reveal subtle ductal irregularities, signal intensity changes, and loss of lobulations, which are more frequently visualized than with CT [1,8]. MRCP criteria for ECP include MPD dilatation (2–4 mm), pseudocysts ≤ 1 cm, and an irregular MPD with ≥3 pathological side branches [21].

When performing MRCP, intravenous administration of secretin may improve diagnostic efficacy in CP. Secretin increases the secretion of pancreatic juice into the MPD, which enhances the visibility of the MPD and its lateral branches, and may help detect stenoses and abnormal extensions of the MPD and its branches. Furthermore, it enables quantitative assessment of pancreatic exocrine function through the analysis of duodenal fluid volume following secretin stimulation [6,20].

A four-grade duodenal filling (DF) score, which determines the extent of pancreatic fluid secretion, has been proposed. Pancreatic exocrine function is considered reduced when the DF score is lower than 3 [6].

### 3.4. Endoscopic Retrograde Cholangiopancreatography (ERCP)

ECP in ERCP is diagnosed based on the criteria of more than three pathological side branches and a normal main pancreatic duct [1]. ERCP enables detailed pancreatography and reveals changes related to chronic fibrosis and atrophy, such as abnormalities of the main pancreatic duct (MPD), calcifications, large cavities, filling defects, strictures, and irregularities in ductal contour [15].

Nevertheless, ERCP is used only for therapeutic interventions due to its high invasiveness and possible complications. This method has the following disadvantages: it is highly operator-dependent, and due to the forceful application of contrast across the pancreatic duct, it may falsely detect ductal changes [15].

### 3.5. Endoscopic Ultrasound

Endoscopic ultrasound (EUS) is an endoscopic imaging technique that visualizes the pancreas through the esophagus, stomach, and duodenum without interference from gas, fat, or bone. There are two categories of EUS: radial and linear. The first enables circumferential views at right angles to the shaft of the scope. Linear EUS visualizes the pancreas in the same plane as the scope, which resembles transabdominal US [22].

According to many studies, EUS has higher sensitivity (84%) for detecting fibrosis in patients with CP compared to other imaging modalities [1,15,23]. It can visualize the side branches and mild contortions of the MPD. Signs of pancreatic fibrosis include hyperechoic foci and strands, parenchymal lobularity, and a hyperechoic ductal wall [1].

In patients with a strong suspicion of CP and negative CT and MRI results, EUS should be performed. EUS can detect very subtle changes in the pancreatic parenchyma and ducts that are not detectable by other imaging methods and functional testing, such as stranding, hyperechoic foci without shadowing, lobularity with or without honeycombing, cysts, dilated side branches, and hyperechoic MPD margin [1].

Among 22 patients with early CP, EUS detected signs of inflammation (narrowing, dilatation, side branch dilatation, irregular contour, parenchymal echogenicity, echogenic foci, cysts) in 19 patients, while ERCP showed no lesions in 11 patients and minimal changes in 11 others. In this study, the sensitivity of EUS was 86%, and for ERCP, it was only 50% [24].

In another study comparing the efficacy of ERCP and EUS in detecting early PDAC, the sensitivity values were 80.7% and 100%, respectively. In this study, patients with clinically improved or suspected CP participated. Positive results of ERCP, recurrent upper abdominal pain, and a history of chronic alcohol use were the inclusion criteria. Early changes detected with EUS included accentuation of the lobular pattern, focal areas of reduced echogenicity, hyperechoic foci (>3 mm diameter), and increased duct wall echogenicity. These patients were then included in a follow-up program that lasted 6–25 months. They were clinically and morphologically examined, and endoscopic examination was performed in cases of increased pain, episodes of acute pancreatitis, or increased pain plus acute pancreatitis. Moreover, 22 of 32 patients with changes in the first EUS examination but normal ERCP developed CP, which was confirmed in ERCP after the follow-up program. Diagnosis of CP was assessed based on Wiersema criteria [25].

On the other hand, EUS may visualize changes in the side branches and mild contortions of the MPD in normal individuals [26]. Side branches exceeding 1 mm are considered abnormal. The overdiagnosis of ECP on EUS is a particular concern in elderly individuals [1]. Age-related changes in the pancreas are similar to those of ECP. Some parenchymal and ductal features, such as fatty replacement or fatty infiltration of the pancreas, acinar atrophy, dilated MPD, parenchymal atrophy, and fibrosis, have been found in elderly people as well as in persons with a history of alcohol abuse, smokers, and patients with other risk factors [9,27,28].

Using EUS-fine needle aspiration (EUS-FNA) for diffuse CP, the negative predictive value increased to 100% compared to 75% for EUS alone; moreover, specificity increased to 67% versus 60%. Thirty-seven patients with suggestive CP based on clinical symptoms and laboratory tests underwent EUS-FNA. The collected samples were subjected to cytological analysis after Papanicolaou staining. Common cytological criteria for inflammation and malignant cellular transformation were assessed. Mild to moderate lymphocytic or granulocytic infiltration, hyperplastic epithelial cells, occurrences of degenerated tissue, and sparse protein precipitates corresponded to mild CP. For severe CP, characteristics included dense infiltration of lymphoid cells or macrophages, massive epithelial degeneration, necrosis or cellular debris, numerous protein precipitates, and calcifications [29]. Table 1 compares the detectability of ECP across various diagnostic methods [1,6,18,30,31,32].

## 4. Rosemont Diagnostic Criteria

The Rosemont criteria (RC) were published in 2009. This classification system attempts to standardize and more explicitly define the endosonographic features for the diagnosis of chronic pancreatitis. The criteria are classified into Major and Minor categories of importance [33]. They are useful for the diagnosis of CP and its staging (mild, moderate, severe), based on the number of conventional EUS findings [13].

The RC is divided into parenchymal criteria and ductal criteria. Among the parenchymal criteria, hyperechoic foci, lobularity, cysts, and strands can be highlighted. Hyperechoic foci are subclassified into two categories, i.e., with shadowing and without shadowing. Features such as MPD calculi, MPD dilatation, irregular MPD contour, dilated side branches, and hyperechoic MPD margin are classified as pancreatic ductal findings. The RC classifies EUS findings as Major A, Major B, and Minor [13].

Major A includes hyperechoic foci with shadowing and MPD calculi. Major B refers to lobularity with honeycombing. The minor criteria comprise lobularity without honeycombing, hyperechoic foci without shadowing, MPD dilatation, irregular MPD contour, cysts, strands, dilated side branches, and a hyperechoic MPD margin [13].

According to the RC, each patient can be classified as consistent with CP, suggestive of CP, indeterminate for CP, or normal [11]. Consistent with CP is recognized when one Major A feature and at least three minor criteria are present. Diagnosis of suggestive CP is based on one Major A finding and fewer than three minor findings or one Major B finding and at least three minor features [27]. If three to four minor criteria or lobularity with honeycombing and fewer than three minor criteria are present, indeterminate CP may be diagnosed. Two or fewer minor criteria, excluding cysts, dilated side branches, MPD dilatation, or hyperechoic foci without shadowing, are recognized as normal [13].

Most EUS features of CP are typically located in the body and tail of the pancreas. It may be difficult to diagnose focal CP in the head, which results only in an enlarged pancreatic head. This appearance of CP may resemble pancreatic cancer. In addition, both of these entities cause fibrosis and ductal obstruction. Sometimes CP is located only in the focally enlarged head of the pancreas while the body and tail are free from inflammatory changes. In rare cases, the inflammatory process can lead to atrophy of the underlying stromal pattern. In cases of solid focal changes, due to PDAC suspicion, the final diagnosis is often made after surgical resection and pathologic examination of the surgical specimen [34].

## 5. Japanese Diagnostic Criteria for Early Chronic Pancreatitis (ECP) 2009

The Japan Pancreas Society (JPS) was the first to publish guidelines for ECP in 2009. According to the JPS, ECP is defined as the absence of typical CP findings and the presence of at least two of the four clinical and imaging findings of ECP on EUS. Clinical findings include repeated epigastric pain, abnormal pancreatic enzyme levels in the serum or urine, impaired pancreatic exocrine function demonstrated by direct tests (cholecystokinin/pancreozymin or secretin test) and indirect tests (fecal elastase-1, fecal fat measurements, mixed triglyceride breath test, and serum trypsinogen), and continuous heavy alcohol consumption (≥80 g/day of pure ethanol) [13,16].

Features characteristic of ECP include lobularity with honeycombing, lobularity without honeycombing, hyperechoic foci without shadowing, stranding, cysts, dilated side branches, and a hyperechoic MPD margin. For an ECP diagnosis, at least two of these seven features must be present [11]. Patients with repeated epigastric pain or elevated amylase and/or lipase levels in the serum or urine, together with EUS findings of ECP, are defined as probable ECP [13].

Clinical findings that do not suggest ECP include pancreatic atrophy, fibrosis, pain syndromes, duct distortion, calcifications, pancreatic exocrine dysfunction, pancreatic endocrine dysfunction, and dysplasia [35].

## 6. Japanese Diagnostic Criteria for Early Chronic Pancreatitis (JDCECP) 2019

The criteria from 2009 have not been accepted internationally. The criteria used so far have been characterized by low specificity, lack of histological correlation, and an uncertain clinical course [11,21]. Therefore, the JPS CP criteria were updated in 2019 to increase specificity and improve usability.

New adjustments have been introduced, such as the inclusion of risk factors and clinical changes [21]. Clinical criteria for CP include repeated epigastric or back pain, abnormal pancreatic enzyme levels in serum or urine, abnormal pancreatic exocrine function, continuous heavy alcohol consumption (≥60 g/day of pure ethanol) or pancreatitis-related susceptibility genes, and a previous history of acute pancreatitis. ECP is diagnosed when at least three of the above features are present. When only two components are present, it is recognized as probable ECP [13].

EUS features include hyperechoic foci without shadowing or stranding, lobularity (either non-honeycombing or honeycombing type), hyperechoic MPD margin, and dilated side branches. If at least two findings, including hyperechoic foci without shadowing or stranding and lobularity, are observed on EUS, ECP may be diagnosed [13].

## 7. Modified JDCECP 2019

EUS findings from the Rosemont Criteria were incorporated into the JDCECP in 2019. The criteria from RC are scored as follows: features from the Major A group are assigned 5 points, findings from the Major B group 3 points, and 1 point is assigned to Minor group findings. A total score of 8 points or more indicates a diagnosis of CP, 6–7 points correspond to CP, 3–5 points are characteristic of ECP, and a score of (at most) 2 points is considered normal [13].

The Rosemont criteria are used exclusively for pancreatic assessment in EUS. Detection of ECP using the above criteria was compared in the study involving 97 patients who underwent EUS. Based on the JDCECP 2009 criteria, ECP was diagnosed in 77 patients. On the other hand, based on the JDCECP 2019 criteria, ECP was recognized in 51 patients. In addition, 42 patients had ECP according to the modified JDECP 2019 criteria, with features from Major A. The modified JDECP 2019 criteria, incorporating Major B from RC, diagnosed 35 cases of ECP [13]. They helped detect the early stages of chronic pancreatitis. Most changes that are typical of ECP are detectable by EUS. Other imaging modalities are less crucial and often inappropriate for detecting ECP because the changes are very subtle and not visible.

## 8. Pancreatic Cancer—Introduction

Pancreatic cancer (PDAC) is one of the deadliest diseases. The Global Cancer Statistics 2022 indicate that the overall incidence of pancreatic cancer was 510,992 cases, with a total global PDAC mortality of 467,409. The mortality rate among PDAC patients is very high, amounting to 91.47%. This makes pancreatic cancer the 12th most common cancer and the 6th leading cause of cancer-related deaths worldwide [36]. According to the WHO, the number of new PDAC cases in Poland in 2022 was 5881, with 5795 deaths reported [37]. Morbidity is higher in males than females and increases gradually after age 45, peaking around age 80. Mortality is highest in the 55–74 age group. Long-term smoking, poor dietary habits, and related conditions such as CP, diabetes, and obesity contribute to the development of PDAC [38]. Despite continuous progress in research and diagnostics, PDAC remains a significant diagnostic and therapeutic challenge [39]. It is estimated that PDAC will become the second leading cause of cancer-related deaths by 2030 [40]. The disease has a poor 5-year survival rate of less than 10% [38,40,41,42,43,44,45]. Additionally, PDAC is characterized by highly aggressive biological behavior. At diagnosis, only 15–20% of patients are eligible for radical surgery [46]. Due to these unfavorable mortality statistics, early PDAC detection is necessary.

Early PDAC presents with non-specific symptoms that rarely alert patients or medical staff. Consequently, most patients are diagnosed at metastatic or locally advanced stages [22]. Pancreatic abnormalities in advanced PDAC include cystic lesions, CP-like parenchymal changes, and solid tumors [42]. Currently, no serum biomarkers are available for early PDAC diagnosis in the general population, unlike those for prostate or ovarian cancers. Various classes of serum biomarker assays—including proteins, autoantibodies, circulating DNA, microRNAs, methylated DNA, and exosomes—have been evaluated for PDAC diagnosis, but none are yet clinically useful [4,7,10,15,43,46].

Despite developments in new drugs, treatment outcomes remain unsatisfactory. Surgical resection combined with chemotherapy only slightly improves survival. Smaller tumors measuring 1–2 cm—detected at an early stage and deemed resectable—have a better prognosis, with a 41% 5-year survival rate [46,47]. According to the American Cancer Society, the 5-year relative survival rate is 44% for localized tumors, 16% for regional spread, and 3% for distant metastases [48]. For tumors limited to the duct epithelium and smaller than 1 cm, the 5-year survival rate can reach 100% [38]. This was confirmed in a follow-up study by Ariyama involving 44 patients, where small tumors were diagnosed using ERCP and angiography. Non-specific changes such as obstruction, stenosis, and irregular cystic dilatation of the pancreatic duct system were visualized during ERCP. In this study, the 5-year survival rate after surgery for tumors smaller than 1 cm was 100%, whereas fewer than 50% of patients with tumors larger than 2 cm survived 5 years [49]. Tumors larger than 2 cm were more often associated with a higher histologic grade, lymph node involvement, advanced T-stage, and a more advanced clinical stage compared to tumors smaller than 2 cm. R0 resection was achieved more frequently in patients with tumors smaller than 2 cm than in those with larger tumors (86% vs. 75.2%) [45]. Table 2 presents the survival outcomes of PDAC patients according to tumor size [50,51,52,53,54].

## 9. Screening in High-Risk Individuals

High-risk individuals (HRIs) for pancreatic ductal adenocarcinoma (PDAC) include patients with a history of familial pancreatic cancer (FPDAC) or pathogenic germline mutations such as STK11, CDKN2A, BRCA1/2, ATM, PALB2, MLH1, MSH2, MSH6, EPCAM, and TP53. For HRIs aged 61–70 years, the incidence rate (IR) is 24 per 100,000 person-years, while for those over 70 years, the IR rises to 60 per 100,000 person-years [55]. Screening aims to detect imaging abnormalities at an early, potentially curable stage, and the likelihood of early detection is higher in HRIs than in the general population [42].

Screening guidelines for HRI have been developed by the Cancer of the Pancreas Screening (CAPS) Consortium and the National Comprehensive Cancer Network (NCCN). CAPS, established in 2010, focuses on improving early PDAC detection and setting screening protocols for HRI groups. Screening generally begins at age 40, or 10 years prior to the age of the youngest affected family member [42].

According to the latest CAPS guidelines, screening starts at age 40 for all carriers of CDKN2A and STK11 mutations, as these patients have a high lifetime risk of PDAC [56]. For individuals with mutations in BRCA1/2, ATM, PALB2, MLH1, or MSH2, screening is recommended only if there is at least one first-degree relative with pancreatic cancer; it should start at age 45–50 or 10 years younger than the youngest relative diagnosed with PDAC [38]. For patients with Peutz–Jeghers syndrome (STK11 mutation), screening also begins at age 40. In those with familial risk but no detected genetic mutation, surveillance should start at age 50 or later [56].

Imaging methods used for screening include endoscopic ultrasound (EUS), magnetic resonance imaging (MRI), and magnetic resonance cholangiopancreatography (MRCP), typically performed annually. If no abnormalities or only non-worrisome findings (such as pancreatic cysts without concerning features) are detected, screening intervals remain at 12 months. When abnormalities that do not indicate immediate surgery are found, screening should be repeated every 3–6 months. Computed tomography (CT) may also be considered [56].

Additionally, fasting blood glucose and/or HbA1c testing can be included in diagnostics, as new-onset diabetes in patients aged 50 or older often accompanies PDAC diagnosis. Screening intervals are adjusted depending on the severity of abnormalities, as follows: for mild or absent abnormalities, 12 months; for more advanced findings—such as solid lesions smaller than 5 mm or of uncertain significance, main pancreatic duct (MPD) strictures, cystic lesions ≥ 3 cm, MPD diameter between 5–9 mm, lymphadenopathy, increased serum CA 19-9, or lesion growth ≥ 5 mm over 2 years—intervals shorten to 3–6 months, provided immediate surgery is not indicated [56].

NCCN recommends that HRI screening be conducted in high-volume centers and include contrast-enhanced MRI/MRCP and/or EUS annually. If abnormalities are found, screening intervals should be shortened based on clinical judgment [57]. The exact recommended age to start screening varies by genetic mutation and is detailed in Table 3 [56,57].

Routine screening for pancreatic cancer in the general population is not recommended due to cost inefficiency. PDAC incidence is relatively low, and screening methods such as EUS and MRI are costly and not widely accessible.

## 10. Imaging Modalities of PDAC

### 10.1. Transabdominal Ultrasonography (US)

PDAC usually appears on the US as a hypoechoic solid mass with ill-defined margins [58]. The use of contrast agents or oral water administration may help visualize the entire organ. Contrast-enhanced US can aid in differentiating pancreatic cancer from other focal lesions, such as neuroendocrine tumors or chronic pancreatitis. After contrast administration, PDAC appears hypoechoic in the arterial phase, whereas CP shows echogenicity similar to that of the normal pancreas [6]. Neuroendocrine tumors are well vascularized, with visible hypoechoic lymph nodes, liver metastases, and hypoechoic primary lesions [6,59]. However, this method does not allow for precise diagnosis of small pancreatic tumors [58]. The US may detect tumors smaller than 1 cm, but sensitivity ranges widely from 17% to 70% [22].

### 10.2. Computed Tomography (CT)

Sensitivity and specificity for detecting PDAC range from 89% to 93% and 87% to 100%, respectively [35,58]. PDAC typically appears as a poorly defined, spiculated, hypoattenuating mass with distal atrophy of the gland [35]. Pancreatic duct dilatation or the double-duct sign, caused by obstruction of the pancreatic and common bile ducts, can suggest PDAC [44,58]. The main limitation of CT is its low sensitivity in detecting early lesions and tumors smaller than 2 cm, with a sensitivity of approximately 42.8% [60]. CT can detect lesions smaller than 1 cm with a sensitivity ranging from 33% to 75% [22]. It is recommended to perform multiphasic contrast-enhanced CT, including late arterial and portal venous phases, to evaluate peripancreatic venous and arterial vascular involvement and assess tumor resectability [47]. Figure 2 presents imaging of PDAC in CT.

### 10.3. Magnetic Resonance Imaging (MRI)

PDAC is usually described as a hypointense focal lesion on fat-suppressed T1-weighted imaging and during the pancreatic parenchymal phase of dynamically enhanced, fat-suppressed, T1-weighted sequences, while its appearance on T2-weighted images is variable. According to some studies, it may appear as hyperintense, isointense, or hypointense [58]. The sensitivity, specificity, and accuracy percentages of MRI for diagnosing PDAC have been reported as 93%, 89%, and 90%, respectively [35]. The detection rate of early PDAC with MRI is 71% [61]. MRI is more effective than CT for diagnosing small tumors, hypertrophy of the pancreatic head, isoattenuating pancreatic cancer, and focal fatty infiltration of the parenchyma [58].

### 10.4. Endoscopic Ultrasound

EUS can detect small solid lesions and additionally enable fine needle aspiration biopsy (FNA), which helps obtain a pathological diagnosis [58]. The sensitivities and specificities of EUS-FNA for diagnosing PDAC were 85–92% and 96–98%, respectively [22].

A study by Maguchi et al. assessed the ability of different imaging modalities to detect PDAC depending on tumor size (*n* = 21). They found that the sensitivities of EUS, CT, and US in detecting tumors smaller than 2 cm were 95.2%, 42.8%, and 52.4%, respectively [38]. According to Canto et al., the sensitivity of EUS for detecting smaller solid lesions (<2 cm) is 93%, compared to 53% for CT and 67% for MRI [42]. In a study by Muller et al., involving 49 patients with clinical suspicion of PDAC, EUS demonstrated higher sensitivity for detecting tumors smaller than 3 cm in diameter compared to other imaging modalities [62]. Another study reported sensitivities of 93%, 53%, and 67% for EUS, CT, and MRI, respectively, in detecting pancreatic tumors smaller than 3 cm [22]. 

Rösch et al. and Glasbrenner et al. proposed EUS criteria for malignant tumors, including signs of invasion of adjacent organs, enlargement of adjacent lymph nodes, and masses with irregular outer margins [2].

EUS provides good visualization of invasion into major peripancreatic vessels, such as the splenic vein, portal vein, and proximal superior mesenteric artery, with an accuracy of 67–93%. The sensitivity and specificity for detecting vascular invasion were 42–91% and 89–100%, respectively. The sensitivity of EUS for detecting tumor invasion of the portal vein exceeded 80% and was better than that of CT [22]. Features indicative of vascular invasion include irregularity of the interface with vessels, intravascular tumor growth, and non-visualization of the vessel with collateral circulation development [2].

EUS also enables more accurate staging than other imaging techniques. When used for staging, the accuracy of T-stage assessment was approximately 78–91%, and the accuracy of N-stage assessment ranged between 41 and 86%. The accuracy of T-stage assessment via EUS was higher than that of CT in patients with tumors smaller than 2 cm [2]. Sensitivities of contrast-enhanced CT and EUS were 50% and 94.4%, respectively [63].

Intraductal papillary mucinous neoplasm (IPMN) is a precursor of PDAC. It is more common in elderly patients and is mainly located in the pancreatic head. Papillary proliferation of the ductal epithelium and dilatation of the excretory pancreatic ducts may indicate IPMN. EUS provides good visualization of the communication between the main pancreatic duct (MPD) and a dilated side pancreatic duct, as well as thickening of the pancreatic duct wall or mural nodules. The sensitivity, specificity, and diagnostic accuracy of EUS for detecting IPMN ranged from 80% to 100%, 78% to 85%, and 79% to 92%, respectively [2,64,65].

Another precancerous condition is pancreatic intraepithelial neoplasia (PanIN), which is characterized by metaplastic and dysplastic proliferation of the ductal epithelium. Over time, as a result of mutations and genetic changes, PanIN can progress to higher degrees of dysplasia and invasive PDAC. There are three grades of PanIN—PanIN-1 and PanIN-2 represent low-grade changes, while PanIN-3 is a high-grade change with high malignant potential. These lesions may cause lobulocentric acinar atrophy by obstructing small ducts with secretions and inducing local inflammation in adjacent tissues [66]. Such precursor lesions may be too small to detect with currently available imaging modalities, but their effects on the pancreatic parenchyma can be identified and quantified—for example, non-uniformly dispersed atrophy may present as a heterogeneous pattern on EUS [67]. Figure 3 presents imaging of PC in EUS. Table 4 compares early PDAC detection results using different imaging techniques [22,28,35,60,68,69,70].

According to the majority of the studies, the sensitivity of EUS in detecting early PDAC was the highest, in some studies as high as 100%.

## 11. EUS-Related Techniques

EUS may be combined with several additional techniques, such as EUS elastography, EUS-guided fine needle aspiration, and contrast-enhanced EUS.

### 11.1. EUS Elastography

Elastography enables assessment of the firmness or elasticity of a target lesion compared to the surrounding normal tissue. The elastography data can be displayed using a three-color system, i.e., red for soft tissue, green for average hardness, and blue for hard tissue [1,43]. Elastography can aid in differentiating between chronic pancreatitis and pancreatic cancer, although it is not definitive. Pancreatic cancer is hard, appearing blue, while inflammatory lesions appear green to yellow. This technique can be particularly useful in patients where distinguishing between inflammatory and cancerous components is challenging [43].

EUS elastography has been reported as a valuable supplementary modality for ruling out malignant pancreatic lesions due to its high negative predictive value. The sensitivity for diagnosing mild or higher-grade fibrosis is 76.4%, with a specificity of 91.7% [1]. In another study involving 68 patients undergoing pancreatic resection with islet autotransplantation for non-calcific chronic pancreatitis, EUS sensitivity without elastography was 61%, and specificity was 75% [44].

Among 68 patients with various pancreatic lesions—including chronic pancreatitis, PDAC, and neuroendocrine tumors—EUS elastography demonstrated sensitivity, specificity, and accuracy in differentiating benign and malignant masses of 91.4%, 87.9%, and 89.7%, respectively [71]. Giovannini et al. compared EUS elastography with conventional B-mode EUS in distinguishing benign from malignant pancreatic masses. Final diagnoses were confirmed through histology obtained from EUS-FNA and/or surgical specimens. EUS elastography differentiated benign from malignant masses with a specificity of 80% and a sensitivity of 92.3% [72].

Other studies reported that EUS elastography had 95–97% sensitivity and 67–76% specificity in differentiating solid pancreatic tumors [73]. 

Kim et al. reported that EUS elastography, using a strain ratio (SR) cut-off value of 5.62, achieved sensitivity, specificity, and accuracy of 71.6%, 75.2%, and 74.8%, respectively, for detecting chronic pancreatitis. In the same study, PDAC detection using EUS elastography showed a characteristic SR cut-off value of 8.86, with sensitivity, specificity, and accuracy of 95.6%, 96.3%, and 96.2%, respectively. In the study by Kim et al., 398 patients without pancreatic disease, 67 patients with chronic pancreatitis (CP), and 90 patients with pancreatic ductal adenocarcinoma (PDAC) underwent EUS and EUS elastography. The aim of the study was to determine cut-off values for distinguishing normal pancreas, CP, and PDAC. Patients with no history of pancreatic disease and no pathological lesions were classified as controls. CP was confirmed in patients presenting clinical symptoms such as abdominal pain or a history of recurrent pancreatitis, along with meeting EUS-based criteria for CP. Diagnosis of PDAC was based on positive cytology obtained by EUS-FNA in unresectable tumors or histology from surgical specimens [74]. Table 5 summarizes the results of PDAC and chronic pancreatitis detection using EUS elastography [71,72,73,74,75,76,77,78].

EUS elastography has shown promising results in detecting small (≤20 mm) PDAC. Kataoka et al. examined 126 patients with previously detected small solid pancreatic lesions using EUS elastography, reporting a sensitivity of 95% and specificity of 53% [79].

While EUS elastography is a valuable technique for evaluating pancreatic lesions, it cannot replace histopathological confirmation and should not be used as a first-line diagnostic tool [72].

### 11.2. EUS-Fine Needle Aspiration (EUS-FNA)

In EUS-FNA, the sample can be a cytology specimen obtained through a hollow needle [43]. This procedure is recommended for patients with borderline resectable disease and locally advanced stage prior to neoadjuvant chemotherapy. Additionally, these examinations are necessary before starting chemotherapy in patients with unresectable tumors or when metastases are present [66]. EUS-guided procedures can also be used to place fiducials in pancreatic tumors or local lymph nodes during stereotactic radiotherapy. Under EUS guidance, fiducials can be precisely positioned even in very small lesions. Nowadays, needles with pre-loaded fiducials are available, allowing placement without removing the needle assembly for reloading, which shortens the overall procedure [43].

Typically, EUS-FNA uses needles of calibers ranging from 19G to 25G for biopsies of pancreatic tumors, enlarged lymph nodes, or hepatic metastases, to confirm diagnosis and determine the tumor stage [22,73]. There are no strict recommendations regarding needle caliber choice. According to Facciorusso et al., there is no significant difference between using 22 G and 25 G needles [80]. Using a linear EUS scope, needles are passed through the stomach or duodenal wall into the target lesion, avoiding vascular structures [43]. Rapid on-site cytologic evaluation (ROSE) aims to improve the diagnostic functions of EUS-FNA. During ROSE, collected samples from EUS-FNA were stained, and a pathologist assessed adequacy on-site. The sensitivity and specificity in diagnosing pancreatic lesions with ROSE were 82.9% and 100%, respectively [81].

EUS-FNA is generally a safe method. Complications such as pancreatitis, infection, bleeding, and pain occur in only 0–2% of cases [2,22]. EUS-FNA has high accuracy for diagnosing pancreatic carcinoma, reported to be between 80% and 95%. It also demonstrates good diagnostic efficacy for PDAC tumors smaller than 1 cm, with specificity, sensitivity, and accuracy ranging from 80–100%, 40–100%, and 47–100%, respectively [82,83,84]. In a study involving 69 patients with chronic pancreatitis and suspected PDAC—a particular diagnostic challenge—EUS-FNA showed higher sensitivity, specificity, and accuracy compared to EUS alone. The sensitivity, specificity, and accuracy of EUS-FNA in differentiating CP from PDAC were 72.7%, 100%, and 95.7%, respectively, while EUS alone showed values of 63.6%, 75.9%, and 73.9%, respectively. Final diagnoses were confirmed by surgery or clinical follow-up [2]. However, the negative predictive value of EUS-FNA was relatively low (64%) [42].

Most guidelines, including those from the Working Group of the Polish Pancreatic Club, recommend EUS-FNA for patients eligible for chemotherapy or chemoradiotherapy, or when the nature of the tumor is unclear. Patients with resectable PDAC do not require a biopsy and should be referred directly for surgery [47,85]. The NCCN recommends a biopsy before neoadjuvant therapy and for patients with locally advanced pancreatic cancer, borderline resectable disease, or metastatic disease [57]. Other organizations, such as the European Society of Gastrointestinal Endoscopy, suggest obtaining tissue samples via EUS when a pathological diagnosis is required [86]. EUS-FNA remains the most commonly used method for pancreatic lesion biopsies due to its high efficacy and low post-procedural complication rates [58].

### 11.3. EUS-Fine Needle Biopsy (EUS-FNB)

In EUS-FNB, a fine core of tissue is acquired using specially designed needles [43]. There are three types of needles, namely, the Franseen needle, the reverse-bevel needle, and the Menghini-tip needle. The Franseen needle features a crown apex with three-plane symmetric cutting edges. The reverse-bevel needle features a side-fenestrated opening on the needle shaft. The Menghini-tip needle has a tapered bevel edge. Their diagnostic accuracy was assessed, with the Franseen needle showing the best results in histologic and cell block evaluations. The sensitivity and specificity values of the Franseen needle were 95.03% and 100%, and 86.09% and 100%, respectively. The Menghini-tip needle had a sensitivity of 82.67% and specificity of 100% for histology-based diagnosis; for cell block diagnosis, sensitivity and specificity were 55.71% and 100%, respectively. Diagnostic performance for histology with the reverse-bevel needle showed a sensitivity of 82.61% and specificity of 100%, and for the cell block, sensitivity and specificity were 71.98% and 100%, respectively [87].

During EUS-FNB, techniques such as wet suction and slow-pull are used. Via a wet suction, saline is introduced to replace the column of air with liquid. During the stylet slow-pull technique, minimal negative pressure is created in the needle, which improves sampling [86]. Facciorusso et al. compared the effectiveness of different techniques for obtaining tissue by EUS-FNB. Diagnostic sensitivity and specificity were 87% and 100%, respectively, for dry suction. With no suction, diagnostic sensitivity and specificity were 72% and 100%, respectively. Using the slow-pull technique, diagnostic sensitivity and specificity were 84% and 100%. The best results were achieved with the modified wet suction technique, which yielded a sensitivity of 93% and a specificity of 100%. In addition, the no-suction technique was worse in terms of sample adequacy than other techniques. Modified wet suction was recognized as the best technique [88]. According to Francesco Crino et al., the wet suction technique achieved a better tissue core procurement rate (71.4%) compared to the slow-pull technique (61.4%). No significant differences were noted in the following subgroups: for solid pancreatic lesions, the rates were 73.3% (wet suction) and 67.1% (slow-pull); for nonpancreatic lesions, the rates were 67.2% (wet suction) and 48.4% (slow-pull) [89]. Macroscopic on-site evaluation (MOSE) is one of the methods that is supposed to improve the collection of material for cytohistological examination. During MOSE, the endosonographer evaluates the collected sample to determine if it is adequate for further examination. The sensitivity and specificity of PDAC diagnoses were 78–85.2% and 100%, respectively [89]. On the other hand, there is another technique called visual on-site evaluation (VOSE). VOSE enables direct classification of EUS-FNB samples to predict the histological adequacy of specimens obtained during examination [90].

EUS-FNB provides core tissue with preserved architecture of desmoplastic stroma and glandular tissues. It may be particularly useful to establish the histological diagnosis of PDAC and to differentiate malignancy from autoimmune pancreatitis (AIP), chronic pancreatitis, pancreatic lymphoma, or tuberculosis. Additionally, it enabled the procurement of tissue for molecular studies [73]. AIP is characterized by dense collar-like ductocentric infiltration consisting of lymphocytes, plasma cells, and eosinophilic cells. The main histological feature of pancreatic tuberculosis is the presence of granulomas. Interlobular, focal, and bridging fibrosis, pseudocysts or retention cysts, and ductal ectasia are also visible in histological specimens [21,65,91,92].

### 11.4. Contrast-Enhanced EUS (CE-EUS)

Contrast agents consist of gas-filled microbubbles encapsulated by a phospholipid or lipid shell. They are administered through a peripheral vein, and the microbubbles receive transmitted ultrasound waves, causing them to either resonate or be disrupted [22]. Commonly used contrast agents today are SonoVue, which contains sulfur hexafluoride, and Sonazoid, which contains perflubutane [39]. The signal detected in the EUS image is minimally affected by artifacts because this technology can detect signals from microbubbles in vessels with very slow flow, without Doppler-related interference [2,22]. Contrast-enhanced EUS (CE-EUS) typically visualizes most pancreatic ductal adenocarcinomas (PDAC) as solid lesions with hypoenhancement [22].

CE-EUS demonstrates higher sensitivity and accuracy compared to conventional EUS (94.5% vs. 83.1% and 84.1% vs. 78.6%, respectively) [38]. It is particularly valuable for confirming diagnoses when lesions are small and EUS-FNA results are inconclusive. CE-EUS performs significantly better in diagnosing small (<2 cm) pancreatic lesions, with sensitivity, specificity, and accuracy ranging from 83.3% to 95%, 83% to 100%, and 88% to 93%, respectively. CE-EUS also outperforms multidetector-row CT and MRI in the early diagnosis of PDAC. In a study by Yamashita, which involved patients with pathologically confirmed PDAC, the sensitivity, specificity, and accuracy of CE-EUS for tumors ≤ 10 mm were 70%, 100%, and 77%, respectively. In comparison, multidetector-row CT showed sensitivity, specificity, and accuracy values of 20%, 100%, and 39% for tumors ≤ 10 mm. MRI’s sensitivity, specificity, and accuracy for tumors 11–20 mm were 73%, 33%, and 68%, and for tumors ≤ 10 mm, the values were 50%, 100%, and 62%, respectively [63,93,94]. Another study reported the sensitivity and specificity of CE-EUS for diagnosing PDAC as 91% and 87%, respectively [42].

CE-EUS is often used to detect both solid and cystic pancreatic lesions and for staging PDAC by evaluating vascular involvement and cystic mural nodules. It can differentiate mural nodules from mucous clots with sensitivity and specificity of 100% and 80–97%, respectively. This is superior to CT (sensitivity 58%, specificity 100%) and conventional EUS (sensitivity 97%, specificity 40%) [23,95].

In a study involving 277 patients with pancreatic solid lesions detected by EUS, CE-EUS results were compared with multidetector-row CT. The study assessed CE-EUS’s ability to differentiate small ductal carcinomas from other solid tumors such as neuroendocrine tumors, inflammatory pseudotumors, metastases, and solid pseudopapillary neoplasms. Lesions were classified based on contrast enhancement patterns, as follows: hypoenhanced lesions were mostly PDAC, hyperenhanced lesions were neuroendocrine tumors, and isoenhanced lesions were associated with inflammatory changes. CE-EUS detected PDAC with a sensitivity and specificity of 95.1% and 89%, respectively. Among 204 PDAC cases, 194 appeared hypoenhanced, 7 were isoenhanced, and 3 were hyperenhanced. For small PDACs (≤2 cm), CE-EUS sensitivity and specificity were 94.4% and 91.2%, respectively, outperforming CT, which showed 70.6% sensitivity and 91.9% specificity. Final diagnoses were based on histology from resected tumors or histology and cytology of EUS-FNA samples [93].

Table 6 presents the results of diagnosing small PDAC using different imaging methods [63,79,82,83,84,93,94].

## 12. EUS and Biomarkers

The biomarker Ca 19-9 has relatively low specificity and sensitivity, reported at approximately 75% and between 41 and 86%, respectively. In a study involving 546 patients with first-degree relatives affected by PDAC, only 4.9% had elevated Ca 19-9 levels. Among those with elevated Ca 19-9, 18.5% had pancreatic lesions detected by EUS, but PDAC was ultimately diagnosed in only one patient [96].

Ca 19-9 gives false-negative results in patients with the Lewis blood type-negative phenotype and false positives in those with obstructive jaundice, chronic pancreatitis, cholangitis, and various other conditions. This limitation arises because Ca 19-9 is secreted by normal epithelial cells of the pancreas, biliary tract, stomach, colon, endometrium, and salivary glands. While Ca 19-9 levels tend to be higher in cancer than in benign conditions, it is not tumor-specific and can be elevated in gallbladder cancer, cholangiocarcinoma, and ampullary cancer. When combined with other tumor markers such as CEA, CA125, and CA242, the diagnostic sensitivity and specificity for PDAC increased to 90.4% and 93.8%, respectively. Many researchers have advocated for the use of tumor marker panels, including Ca 19-9, to improve PDAC detection [38,97].

CEA is also elevated in both PDAC and benign pancreatic diseases like chronic pancreatitis, with a positive predictive value (PPV) of 77%, a negative predictive value (NPV) of 95%, and an accuracy of 85% for pancreatic cancer diagnosis [61].

Emerging biomarkers under investigation for early detection of preinvasive pancreatic intraepithelial neoplasia include plectin-1, PAM4 antibody, trefoil factors, MUC5AC, apolipoprotein isoforms, and circulating RNA [3,97,98,99]. Among these, inflammatory markers such as IL-6, IL-8, IL-10, IL-12, TNF-α, ICAM, fractalkine, and certain adipokines show promise as PDAC biomarkers. Additionally, markers associated with fibrosis, including MMPs, TIMP-1, hyaluronic acid (HA), TGF-β, PDGF-BB, and MCP-1, are being evaluated. However, none of these markers have yet been incorporated into routine clinical practice [100].

## 13. Diabetes as an Early Indicator of PDAC

Patients with diabetes lasting more than five years have a 1.5–2 times increased risk of developing pancreatic ductal adenocarcinoma (PDAC), whereas those with diabetes of less than five years’ duration have a 5–8 times higher risk. The loss and dysfunction of pancreatic β-cells in diabetes are attributed to oxidative stress, apoptosis, aging, and reductions in β-cell mass caused by malignancy and inflammation. Notably, PDAC may manifest as early as 2–3 years after the diagnosis of new-onset diabetes [101,102].

Diabetes is significantly more prevalent in patients with PDAC compared to healthy individuals or those with other cancers. In a study by Aggarwal et al., 40.2% of patients with PDAC were diagnosed with diabetes within 36 months prior to cancer diagnosis, compared to only 20.5% of patients with lung, breast, prostate, or colorectal cancers [103]. Furthermore, new-onset diabetes is associated with decreased survival in PDAC patients.

Diabetes is frequently accompanied by obesity, an additional established risk factor for PDAC. Both conditions contribute to increased insulin resistance, involving alterations in insulin receptor signaling and glycogen synthesis. In the skeletal muscle, the primary site of insulin resistance, nonoxidative glucose disposal via glycogen synthesis, is impaired due to limited glycogen synthase activity. Insulin normally activates glycogen synthase through dephosphorylation, but in PDAC patients with diabetes, there is an abnormal skeletal muscle response to insulin stimulation [7].

Type 3c diabetes, which is secondary to pancreatic disease, often precedes the appearance of radiologically detectable PDAC lesions, suggesting it could serve as an early biomarker for PDAC [102,103]. A large comparative study that involved 512 patients with newly diagnosed PDAC and 933 control subjects found that diabetes was present in 47% of PDAC patients—74% of whom had new-onset diabetes—compared to only 7% in the control group. Distinguishing between type 2 diabetes and type 3c diabetes related to PDAC remains a clinical challenge, but may be key to identifying high-risk individuals for targeted screening. Currently, there are no formal guidelines for PDAC screening in patients with new-onset diabetes. However, clinical features such as unexplained weight loss and worsening glycemic control—both atypical for type 2 diabetes—should raise suspicion of underlying PDAC [18,101,104].

## 14. Why Early Diagnosis of CP and PDAC Should Be Implemented

The progression of chronic pancreatitis (CP) is often associated with significant and debilitating complications. Early diagnosis plays a critical role in preventing these outcomes and improving the quality of life of patients. Moreover, CP is a recognized risk factor for pancreatic ductal adenocarcinoma (PDAC); thus, early detection of CP may also contribute to reducing the risk of PDAC development [40].

Evidence from animal models suggests that early chronic pancreatitis (ECP) may be a reversible condition, particularly when the underlying inflammatory stimulus is eliminated. In one such study, prolonged hyperstimulation with cerulein induced recurrent pancreatic injury, ultimately leading to fibrosis. Histological analysis—using collagen quantification and staining—demonstrated significant fibrosis in pancreatic tissue. However, upon cessation of cerulein administration, the pancreatic tissue regenerated, and the fibrosis resolved. These findings indicate that pancreatic fibrosis, especially in its early stages, may be reversible [105]. Another study by Vonlaufen et al. provided additional support for the reversibility of CP. In this study, rats were fed a liquid diet containing ethanol for 10 weeks and were also administered intravenous lipopolysaccharide (LPS) to exacerbate inflammation and disease progression. The combined effect of alcohol and LPS led to acinar cell atrophy and pancreatic fibrosis. When alcohol was withdrawn for either 7 days or 3 weeks, partial regression of fibrosis was observed as early as 3 days, and complete resolution occurred within 7 days. In contrast, continued alcohol exposure resulted in persistent pancreatic fibrosis [106].

These experimental findings highlight the potential for reversing fibrosis in ECP, particularly through the removal of causative agents such as alcohol or inflammatory stimuli. This underlines the value of early intervention and lifestyle modification in managing CP and preventing its long-term complications, including the development of PDAC.

Patients with CP experience progressive impairments in insulin secretion and β-cell dysfunction [107]. The development of endocrine pancreatic insufficiency (EPI), along with pancreatic fibrosis, contributes to the destruction of pancreatic islets. Additionally, disruption of the enteroinsular axis plays a significant role in the pathogenesis of diabetes in CP. Normally, ingested nutrients stimulate the release of incretin hormones such as glucagon-like peptide-1 (GLP-1), secreted by L cells in the ileum and large intestine, and glucose-dependent insulinotropic polypeptide (GIP), secreted by K cells in the small intestine. Both hormones enhance insulin secretion in response to food intake. In CP, EPI leads to impaired nutrient digestion, which in turn blunts the incretin response and reduces postprandial insulin secretion. Moreover, patients with diabetes secondary to CP often exhibit nutritional deficiencies, including reduced levels of zinc and selenium, and decreased linoleic acid due to fat malabsorption. These deficiencies may accelerate atherogenesis. Insulin deficiency further disrupts amino acid transport, resulting in elevated plasma amino acid concentrations. Concurrent glucagon deficiency can also contribute to hyperaminoacidemia. Pancreatic enzyme replacement therapy (PERT) has been shown to improve nutrient digestion and absorption, thereby enhancing incretin hormone secretion and aiding glycemic control [102,108]. Early detection of pancreatic ductal adenocarcinoma (PDAC), especially when associated with endocrinopathies such as insulin resistance, may enable prompt therapeutic interventions and potentially delay or prevent the onset of diabetes. Pharmacologic treatments for these metabolic disturbances include metformin and thiazolidinediones. Metformin reduces hepatic gluconeogenesis and glucose production while increasing insulin sensitivity. Thiazolidinediones also improve insulin sensitivity and reduce hepatic glucose output [109]. Endocrine pancreatic insufficiency is commonly observed in CP patients and tends to be more severe in those with alcohol-induced disease [110].

EUS was performed in 1157 patients with suspected pancreatic disease, 73% of whom were smokers and alcohol consumers. The aim of this study was to evaluate the influence of alcohol and cigarettes on the severity of pancreatic lesions. A history of more than 40 pack-years of smoking was present in 9.4% of the total patients; 39.4% had 5–40 pack-years, and 51.2% had ≤5 pack-years. An ethanol index (number of standard drinks consumed per week × number of years of alcohol consumption at that level) greater than 400 was present in 13.8% of the total patients; 20.8% had an ethanol index of 40–400, and 65.4% had an ethanol index ≤ 40. Moreover, 26% had at least five typical CP imaging changes, such as hyperechogenic foci and strands, lobularity, irregular pancreatic duct margins, and hyperechoic pancreatic duct margins. The presence of five or more features is considered indicative of severe pancreatic abnormalities. Heavy alcohol consumption and a long smoking history may cause severe pancreatic lesions. Stones are independent predictors of heavy ethanol consumption in imaging techniques, and hyperechoic strands are associated with smoking [111]. Smoking and alcohol cause oxidative stress, leading to necrosis and apoptosis of pancreatic cells. Next, activation and infiltration of inflammatory cells occur, which lead to fibrosis [100]. Most of these changes may disappear when the risk factors subside [74]. Possible therapies that may reverse ECP are not evidently established. Antioxidants, TGF-beta suppression, and TNF-alpha inhibition have been proposed as possible treatments for preventing pancreatic ductal cell fibrogenesis [112]. Animal studies show that such interventions may reduce local and systemic inflammatory responses and even decrease mortality rates.

TNF-alpha is a pro-inflammatory cytokine, and its serum concentration has been shown to correlate with the severity of pancreatitis in both animal models and humans. In cerulein-induced pancreatitis in rats, following TGF-beta antibody injection, the deposition of fibronectin and collagen in pancreatic specimens was reduced compared to the saline control group. In addition, pancreatic tissue hydroxyproline concentration was measured, reflecting the collagen content. The results showed that pancreatic hydroxyproline content, as well as the concentration of collagen and fibronectin, were reduced in animals treated with the TGF-beta antibody [59,113].

In another study involving rats with sodium taurocholate-induced pancreatitis, pentoxifylline (a TNF-alpha inhibitor) and oxypurinol (an inhibitor of xanthine oxidase) were administered to inhibit oxidative stress. After treatment, pancreatic samples were taken, and lipase activity was measured. Oxypurinol administration inhibited plasma xanthine oxidase activity, which was measured in plasma using fluorometry. Pentoxifylline administration prevented the rise of TNF-alpha levels in plasma compared to control animals. Histological studies showed that oxypurinol and pentoxifylline, administered separately, partially reduced interstitial edema. Treatment with both drugs together prevented all the inflammatory features, including inflammatory infiltration in the areas adjacent to acinar necrosis [113].

In humans, the prevention of inflammation depends on the cause of pancreatitis. In gallstone pancreatitis, ERCP may prevent recurrent attacks through pancreatic duct stone removal and stricture dilatation. Some reports suggest that lowering plasma levels of apolipoprotein C-III (APOC3) reduces plasma triglycerides and, furthermore, the risk of TG-associated pancreatitis [114,115,116]. It has also been reported that pentoxifylline may reduce TNF-alpha levels in humans. A significant reduction in TNF-alpha release from monocytes was observed, compared to the control group, in subjects treated with oral pentoxifylline for seven days [117].

According to Whitcomb, when ECP is detected, a low-fat diet and fat-soluble vitamins (A, D, E, K) should be implemented. Multiple small meals may minimize pancreatic exocrine stimulation. Exercise, yoga, and other relaxation techniques may increase the quality of life of patients [118]. These can reduce the number of painkillers taken and improve weight gain [119]. Yoga can increase muscle strength and flexibility, improve circulation in the joints and muscles, and stimulate the brain to produce painkilling chemicals. Furthermore, when performing exercises, patients divert attention to other parts of the body and do not focus on pain. In a study examining the impact of yoga on quality of life in patients with CP, the SF-36 index (including bodily pain) increased in the yoga group compared to the control group [38]. Abstinence from smoking and alcohol is suggested. Whitcomb, based on animal models and human studies, showed that prolonged high alcohol consumption and smoking cause adaptive changes, metabolic stress, and immune modulation, which increase CP risk and lead to its development. Substances contained in cigarettes (nicotine) cause pathological and functional changes in the exocrine function of the pancreas, elevating intracellular calcium release and worsening pancreatic blood flow [120]. Smoking causes inflammation, tissue damage, insulin resistance, reduced insulin secretion, endothelial dysfunction, impaired lipid and glucose metabolism, and beta-cell toxicity. These changes dysregulate the exocrine function of the pancreas [121]. The risk of pancreatic calcification increases depending on the pack-year number [122].

If exocrine pancreatic insufficiency (EPI) is diagnosed early in the CP course, pancreatic enzyme replacement therapy (PERT) should be implemented. EPI causes malabsorption, malnutrition, weight loss, steatorrhea, fat-soluble vitamin deficiencies, as well as deficiencies in other vitamins, essential fatty acids, amino acids, and trace elements [108,123]. Another consequence of EPI is osteopathy. A total of 120 PDAC patients were examined to determine the occurrence of EPI and osteopathy. Among them, up to 95% had EPI and 72.1% had osteopathy. Osteopathy was present in 3 patients with ECP, 29 patients with definite CP, and 43 patients with advanced CP. It was shown that in advanced CP, PERT enables weight gain, improves fat absorption, and augments fat-soluble vitamin and trace element levels. In addition, it lowers the risk of bone fracture due to osteoporosis, improves quality of life, and reduces mortality [108]. Moreover, PERT improves quality of life and reduces pancreatic disease symptoms [123].

The latest NCCN guidelines suggest that resectable disease should be treated with surgery or neoadjuvant therapy (NT), which includes chemotherapy (NCT) or chemoradiation (NCR) followed by surgery. Using NT in resectable PDAC is still controversial. According to a study by Takahashi, NT improves median and overall survival in resectable PDAC. Furthermore, more patients can receive all necessary multimodality therapy (a combination of surgical resection and systemic chemotherapy) if NT is implemented [45].

## 15. Conclusions

CP is associated with complications such as progressive nutrient maldigestion, glucose intolerance, diabetes mellitus, and metabolic derangements. Compared to other imaging methods, EUS offers greater sensitivity and specificity for diagnosing ECP. US and CT have limited roles in diagnosing the early stages of CP as they do not detect subtle changes associated with ECP.

EUS can detect PDAC in the early stage in HRI patients when there is a slightly better chance for radical operation and longer survival. EUS has better results in the detection of small tumors (<2 cm) compared to other imaging modalities, which have a better prognosis than advanced lesions. Nevertheless, EUS has some limitations. This imaging modality is highly operator-dependent, and the findings may also have inter-observer variability, affecting the overall accuracy of the diagnosis. Furthermore, it is associated with some risks of the procedure and sedation. In addition, in cases where the anatomy is distorted or surgically altered, revealing optimal imaging may not be possible.

On the other hand, many factors, such as aging, smoking, obesity, and chronic alcohol consumption, may cause EUS changes that are typically seen in CP. There is also a high likelihood of false positives due to age-related physiological changes in the pancreatic duct, which can lead to overdiagnosis of ECP.

EUS seems to be the best modality for identifying small solid lesions and plays a valuable role in differentiating benign from malignant pancreatic masses. It can detect early asymptomatic precursor cancer lesions (e.g., IPMN and PanIN) as well as early invasive malignant pancreatic neoplasms in HRIs.

Additionally, techniques such as CE-EUS, EUS elastography, and EUS-FNA help detect pancreatic lesions with better sensitivity and specificity than EUS alone. EUS-FNA enables diagnosis and histopathological confirmation, as well as contributes to the differentiation of different pancreatic masses. Detecting ECP and PDAC may prevent severe disease complications (such as diabetes and malnutrition), slow down the carcinogenesis process, and significantly improve the patient’s health-related quality of life.

## Figures and Tables

**Figure 1 cancers-17-01891-f001:**
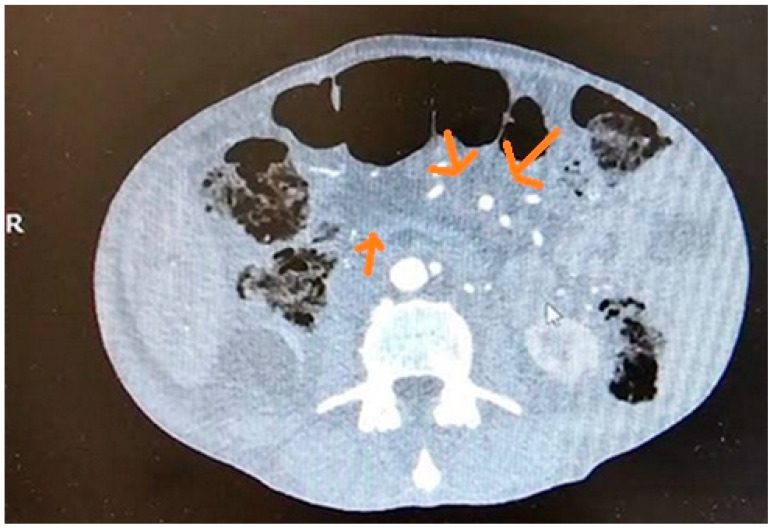
Early chronic pancreatitis in CT.

**Figure 2 cancers-17-01891-f002:**
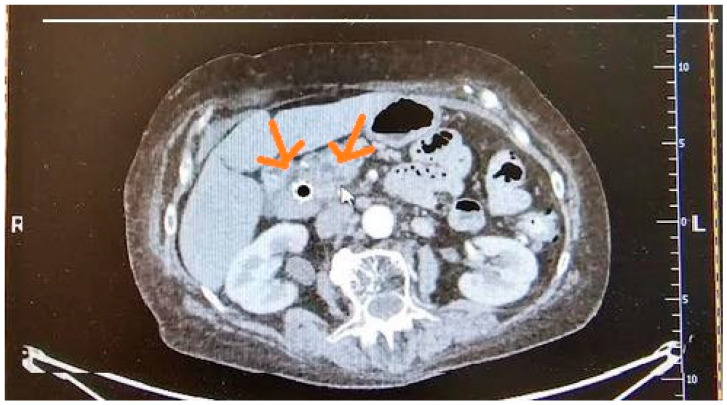
Pathologic masses of pancreatic cancer in CT.

**Figure 3 cancers-17-01891-f003:**
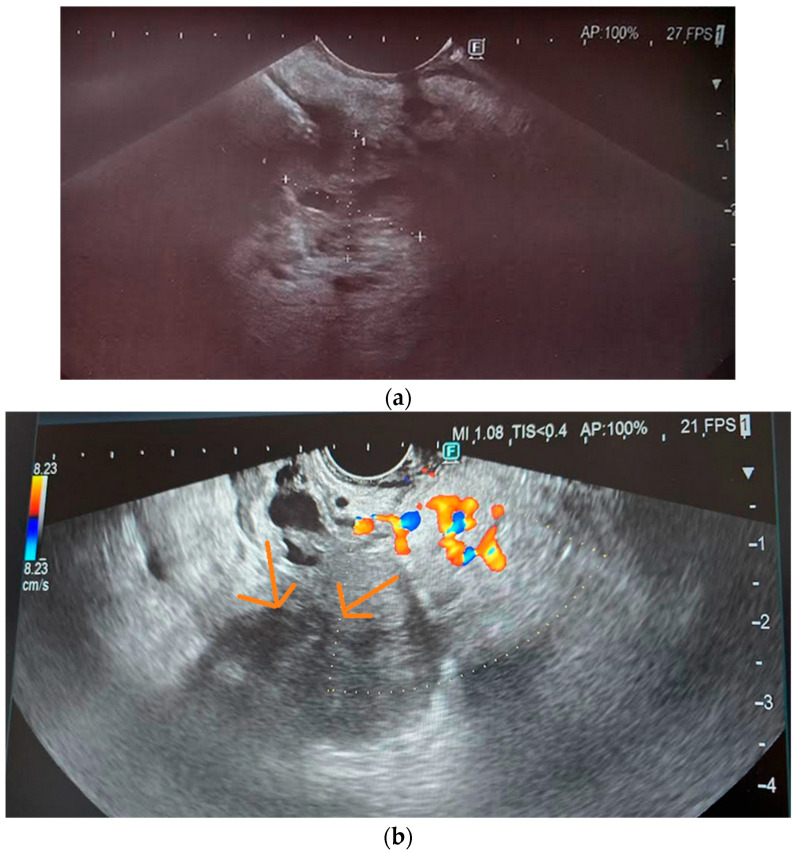
(**a**,**b**) Pancreatic cancer in EUS.

**Table 1 cancers-17-01891-t001:** Comparison of the efficacy of detecting ECP using different imaging methods [1,6,18,30,31,32].

Imaging Modalities	Sensitivity	Specificity	References
US	67–69%	90–98%	[18,30,31]
CT	75%	91%	[18]
MRI	77–78%	83–96%	[18,32]
EUS	81–84%	90–100%	[1,6,8]

**Table 2 cancers-17-01891-t002:** Survival rate in patients with PDAC according to tumor size [50,51,52,53,54].

Size	1 Year	3 Years	5 Years	References
≤2 cm	66.7%	66.7%	26.4–69.9%	[50,51,52]
2.1 cm–5 cm	38.5–81.8%	7.7–36.8%	7.7–24.5%	[51,52,53,54]
>5 cm	64.3%	36.7%	0–36.7%	[52,53]

**Table 3 cancers-17-01891-t003:** NCCN and CAPS guidelines for pancreatic cancer screening [56,57].

Mutation	NCNN Guidelines Age at Screening Start (Years)	CAPS Guidelines Age at Screening Start (Years)
Peutz–Jeghers syndrome (STK11)	30–35	40
Familial pancreatic cancer (CDKN2A)	40	40
Familial pancreatic cancer (BRCA1/2), ataxia–telangiectasia (ATM), PALB2, Lynch syndrome (MLH1, MSH2, MSH6)	50	45–50
Lynch syndrome (EPCAM), Li–Fraumeni syndrome (TP53)	50	-

**Table 4 cancers-17-01891-t004:** Comparison of early PDAC detection efficacy with the different imaging methods [22,28,35,60,68,69,70].

Imaging Modalities	Sensitivity	Specificity	References
US	17–70%	-	[22,68]
CT	33–80%	-	[22,60]
MRI	52.3–93%	77.1–89%	[28,35,69]
EUS	72–100%	90%	[60,68,69,70]

**Table 5 cancers-17-01891-t005:** The efficacy of elastography-assisted EUS in pancreatic solid lesion detection [71,72,73,74,75,76,77,78].

Pathology	Sensitivity	Specificity	Research
Detection of CP	66–72%	57–75.2%	[74,75]
Detection of PDAC	95.6–100%	67–96.3%	[72,74,76]
Differentiation of benign and malignant masses	80–97%	67–92.3%	[71,72,73,77,78]

**Table 6 cancers-17-01891-t006:** Comparison of EUS-FNA, CE-EUS, and CT in the diagnosis of small PDAC (<2 cm) [63,79,82,83,84,93,94].

Imaging Modality	Sensitivity	Specificity	Accuracy	References
EUS-FNA	40–100%	80–100%	47–100%	[82,83,84]
CE-EUS	70–95.1%	83–100%	77–100%	[63,93,94]
EUS elastography	95%	53%	-	[79]
CT	20–70.6%	91.9–100%	-	[93,94]
MRI	50–73%	33–100%	62–68%	[94]

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
