# Peer review of "Changes Connected to Early Chronic Pancreatitis and Early Pancreatic Cancer in Endoscopic Ultrasonography (EUS): Clinical Implications"

_cancers, 2025, doi:10.3390/cancers17111891_

Round 1
Reviewer 1 Report
Comments and Suggestions for Authors
The review was nicely written and it seems quite comprehensive
Some CT or EUS images would improve the quality of the manuscript
The authors should comment the main validation studies of Rosemont criteria
When talking of EUS-FNA, the authors should cite the main evidence on the size of the needle (whether 22G or 25G, in this regard cite the meta-analysis PMID: 29026598 )
When talking of EUS-FNB the authors should comment on the best strategies to achieve good samples (in this regard cite the recent NMA: PMID: 36657607 )
Author Response
We would like to thank the Reviewer for providing feedback to our manuscript. We have gone over the comments and suggestions of the reviewer have revised the manuscript accordingly, which has greatly improved the quality of the manuscript. We added 3 images of CT and EUS and we attached information from proposed articles.
Reviewer 2 Report
Comments and Suggestions for Authors
The review is well balanced and comprehensive although i am not sure the topic completely fit to the scope of the journal.
Some images more could be of help for the manuscript.
The authors should comment more on the techniques to improve the diagnostic yield of EUS-FNB, for example the role of suction or wet techniques (cite the landmark RCT PMID: 35915956 )
Some additional comments on the role of ROSE, MOSE, and VOSE, particularly in light of the recent ESGE guidelines and technical review on tissue sampling, would be of help
Author Response
We agree with the reviewer's comments . As suggested by the reviewer we have added images to the manuscript (FIGURE 1, 2 and 3). In addition we have added new information about techniques that are used in EUS-FNB and the role of ROSE, MOSE, VOSE.
Reviewer 3 Report
Comments and Suggestions for Authors
This manuscript provides a comprehensive and well-referenced overview of the role of EUS in early detection of chronic pancreatitis and pancreatic ductal adenocarcinoma. However, several improvements are needed to enhance clarity, accuracy, and flow:
1.Reduce redundancy: There is frequent repetition of sensitivity/specificity statistics across sections. Consider summarizing comparative values in a single consolidated table or section.
2.Clarify structure: The flow between sections is sometimes abrupt. Use clearer transitions between topics (e.g., from CP to PDAC, or from diagnostic imaging to screening).
3.Improve citation clarity: Some statistics are presented without proper attribution. Ensure that each data point includes a clear reference.
4.Balance depth: The discussion of some imaging techniques (e.g., CE-EUS, EUS-elastography) is highly detailed while others are relatively brief. Consider harmonizing detail level.
5.Refine tables: Tables 2, 5, and 7 are useful but would benefit from clearer formatting and consistent column labels.
6.Scientific tone: Phrases such as “this is very important” or “this indicates clearly” can be revised for more objective, neutral language.
7.Language polishing: Extensive grammatical and syntactical revision is required
Comments on the Quality of English LanguageThe manuscript requires moderate to extensive language editing. Common issues include:
Incorrect or awkward sentence structure.
Misuse of prepositions, articles, and plural forms.
Unnatural phrasing that impairs readability.
Inconsistent terminology (e.g., “ECP” vs. “early CP”).
Professional English editing is strongly recommended to improve clarity and academic tone.
Author Response
Comment 1
We thank the reviewer for pointing out these issues. In the revised manuscript, we have made new table (TABLE 6) which compare diagnosing of PDAC by different imaging methods.
Comment 2
We would like to thank the Reviewer for all comments and suggestions. The structure of the article is that we first describe CP and all related topics, so diagnostic and, RC too. Then we move on to PDAC and describe diagnostics, screeing, etc. In the section about techniques of EUS they can be noticed CP and PDAC together because some researches assess differentiating of CP and PDAC.
Comment 3
We thank the reviewer for pointing out these issues. The manuscript has been revised accordingly.
Comment 4
We have added new information about EUS-FNA and EUS-FNB. In addition divided subsection about EUS-FNA and EUS-FNB in two. Currently section about techniques of EUS have similar length.
Comment 5
We fully agree with the reviewer's comments and we entered column labels to the tables.
Comment 6
We agree with the reviewer and we deleted phrases such as “this is very important” or “this indicates clearly” and revised for more objective, neutral language.
Comment 7
We apologize for not extensive grammatical text. As suggested by the reviewer the manuscript has been edited for linguistic and corrected. We corrected and edited incorrect or awkward sentence structure, misuse of prepositions, articles, and plural forms, unnatural phrasing that impairs readability, inconsistent terminology (e.g., “ECP” vs. “early CP”). We changed early CP for ECP.
Round 2
Reviewer 1 Report
Comments and Suggestions for Authors
The manuscript is OK now. Thank you!
Reviewer 2 Report
Comments and Suggestions for Authors
The authors consistently improved the paper. I have no further comments.